# Comparison of the Ability of Anthropometric Indices to Predict the Risk of Diabetes Mellitus in South African Males: SANHANES-1

**DOI:** 10.3390/ijerph19063224

**Published:** 2022-03-09

**Authors:** Machoene D. Sekgala, Ronel Sewpaul, Maretha Opperman, Zandile J. Mchiza

**Affiliations:** 1School of Public Health, University of the Western Cape, Bellville 7535, South Africa; zandile.mchiza@mrc.ac.za; 2Human and Social Capabilities, Human Sciences Research Council, Cape Town 8000, South Africa; rsewpaul@hsrc.ac.za; 3Department of Biotechnology and Consumer Science, Cape Peninsula University of Technology, Cape Town 7535, South Africa; oppermanm@cput.ac.za; 4Non-Communicable Diseases Research Unit, South African Medical Research Council, Tygerberg, Cape Town 7505, South Africa

**Keywords:** diabetes mellitus, body mass index, waist-to-hip ratio, waist circumference, waist-to-height ratio, South Africa

## Abstract

This study aimed to assess the sensitivity of body mass index (BMI) to predict the risk of diabetes mellitus (DM) and whether waist circumference (WC), waist-to-hip (WHR) and waist-to-height (WHtR) ratios are better predictors of the risk of DM than BMI in South African men aged 20 years and older. Data from the first South African National Health and Nutrition Examination Survey (SANHANES-1) were used. Overall, 1405 men who had valid HbA1c outcomes were included. The sensitivity, specificity, and optimal cut-off points for predicting DM were determined using the receiver operating characteristic (ROC) curve analysis. A total of 34.6% percent of the study participants were overweight/obese, while 10.5%, 10.4%, 36.6% and 61.0% had HbA1c, WC, WHR and WHtR above the normal reference ranges, respectively. Based on age-adjusted logistic regression analysis, the highest likelihood of DM was observed for those participants who had increased WC and WHtR (odds ratios [OR] were 6.285 (95% CI: 4.136–9.550; *p* < 0.001) and 8.108 (95% CI: 3.721–17.667; *p* < 0.001)). The ROC curve analyses for WC, WHR, and WHtR displayed excellent ability to predict the risk of DM, with their areas under the curve (AUC) being 80.4%, 80.2% and 80.8%, respectively. The overall cut-off points to predict the risk of DM for WC, WHR, and WHtR were ≥88.95 cm, ≥0.92, and >0.54, respectively. The ROC analysis for BMI, on the other hand, showed acceptable ability to predict the risk of DM (AUC = 75.6%), with its cut-off point being ≥24.64 kg/m^2^. Even after stratifying the data by two age groups, WHtR remained a superior index to predict DM, especially in the younger age group. To conclude, no significant differences were observed between the AUC for BMI the AUCs for other indices. However, the AUCs for these indices showed significant excellent ability as opposed to the significant acceptable ability of BMI to predict DM in adult South African men.

## 1. Introduction

The prevalence of DM (HbA1_c_ higher than 6.5%) in South Africa increased in the past two decades [1,2,3]. A body mass index higher than 24.9 kg/m^2^ is often regarded as the main anthropometric contributor to the increase in the prevalence of DM in most South African studies [4,5,6]. However, BMI does not indicate body fat distribution [4]. Body fat distribution is a better indicator for the risk of insulin resistance, where insulin resistance is a precursor to DM [7]. Hence, other indices that show body fat distribution such as WC, WHR and WHtR are commonly known as the preferred indicators to predict the development of DM [8,9,10].

In South Africa, the age-standardized prevalence of DM among adults in 2012 was shown to be 10.1% [11]. More recent estimations by the International Diabetes Federation [12] indicated that approximately 4.6 million people between 20 and 79 years old are living with DM in South Africa, representing almost 13% of the total adult population. South African researchers have projected that the prevalence of DM will steadily increase in the near future [3,9,11,12,13,14]. While the aforementioned South African studies show that the prevalence of uncontrolled DM is higher in females, there is concern that if not tightly monitored, the proportion of males who are living with DM may surpass females in future. These predictions are based on current statistics that suggest that there are more males than females (66% versus 64%) who are pre-diabetic in South Africa [1].

Alongside the projected increase in DM in South Africa is the escalating prevalence of individuals with a BMI higher than 24.9 kg/m^2^ [1,2]. In fact, the increase in overweight and obesity in South Africa raises concern, given that elevated BMI is an important risk factor for many of non-communicable diseases (NCDs) including DM [15,16,17]. According to Patel et al. [18], BMI was significantly associated with DM status for all United States cohorts in the National Health and Nutrition Examination Surveys.

There is also growing evidence to suggest that the prevalence of adiposity among South African men may be underestimated by only utilizing BMI [2]. For instance, unlike in women where the prevalence of overweight and obesity, as measured by BMI, has increased by more than 12% since 1998, the prevalence of overweight and obese men has increased by just 2% (from 29% to 31%) during the same period [1,2,14,19]. Despite this low increase, Joubert et al. (2007) [17] reported more than 10 years ago that 87% of South African men older than 30 years, who were not classified as overweight or obese, presented with DM [17]. In most parts of the world, including South Africa, men present with poor ill-defined cardiovascular disease outcomes compared to women [20,21]. This might be attributed to the finding that more men actively smoke [20,21] as well as higher alcohol consumption among men compared to women. Hence, more ambitious NCD risk detection mechanisms may be required for men.

Current international evidence shows that although BMI has been widely used as a measure of obesity [4,22], this index has a significant limitation in that it does not reflect body fat distribution. Moreover, while BMI is a simple and convenient measure for adiposity in many epidemiological studies, its validity and ability to accurately measure adiposity have been questioned as it does not directly measure the amount of adipose tissue and cannot differentiate between body fat and lean mass [23]. There is substantial evidence to suggest that metabolic complications associated with obesity are more closely associated with visceral adiposity than overall body adiposity [24]. As such, other measurements of visceral adiposity, such as WC, WHR and WHtR are widely advocated [10,24]. Visceral adiposity can promote a cascade of secondary risks for cardiometabolic conditions such as hypertension, insulin resistance, hyperuricemia and hyperlipidemia [25]. Some studies have proposed the individual use of WC, WHR or WHtR to measure the aforementioned disease risks [5,26,27] whereas others advocate their combined use [23,28].

With the preceding international evidence that implicates adiposity in the development of DM, to our understanding, no South African study has previously investigated the specificity and sensitivity of BMI to predict DM. In fact, no South African study considered identifying better-performing anthropometric indices to predict the risk of DM. Hence, this study aimed to assess the sensitivity and specificity of BMI to predict DM among South African males aged 20 years and older. Furthermore, this study compared the power of BMI to predict the risk of DM against other indices such as WC, WHR, and WHtR.

## 2. Materials and Methods

### 2.1. Sampling Procedure

This research forms part of a larger analyses of data from individuals who participated in the SANHANES-1 study. The SANHANES-1 was a cross-sectional survey undertaken in 2012 to determine the nutrition and health status of the South African population. The sample size was determined by the requirement of an acceptable precision of estimates per reporting domain; that is, to be able to estimate the prevalence of a given health or nutrition variable as well as societal risk factor(s) in each of the main reporting domains with an absolute precision level of less than 5%, which is equivalent to the expected width of the 95 percent confidence interval (z-score Z_1–α/2_ at the 95% level). A design effect of 2 was assumed to account for possible intra-class correlation. The total sample size of 10,000 households was determined using the minimum sample sizes required for each reporting domain, as well as the multistage cluster sampling design and expected response rates. Assuming that 75% of the 10,000 households in the sampling frame agree to participate, the survey would yield 7500 valid contactable households with eligible survey participants. The average household size demonstrated in the 2008 national HIV household survey [29] was 3.9 people per household, and this figure was used to calculate the expected sample size of eligible individuals by age group for the total sample.

The survey used a stratified cluster sampling approach with multiple stages. A total of 1000 census enumeration areas (EAs) mapped using aerial photography in 2007 were used to produce the Master Sample. The EAs were chosen based on their province and locality type. From the Master Sample, 500 EAs were chosen to represent the socio-demographic profile of South Africa. Random samples of 20 visiting points (VPs) were chosen at random from each EA, providing a total sample of 10,000 homes. As shown in Figure 1, the final sample consisted of 8166 valid and occupied households, with 27,580 eligible individuals of all ages. Of the eligible individuals, 92.6% (25,532) participated in the interviews, 43.6% (12,025) volunteered to undergo a physical examination where anthropometric measures were obtained and 29.3% (8078) donated a blood sample for biomarker analysis.

Of these individuals 17% had missing data on HbA1c and therefore were excluded. A further screening of completeness of data on weight, height, waist and hip circumferences was undertaken. Additional exclusion of those with missing data on the aforementioned indicators was undertaken, resulting in a total sample of 4083 males and females. Of these individuals, because the focus of this research was on males, females were excluded and all males aged 20 years and older (41.5% [N = 1405]) were included in the current analyses. A summary flow diagram of participant selection for this study is presented in Figure 1. Additional information about the SANHANES-1 methodology, content, and laboratory procedures can be found elsewhere [2,11].

### 2.2. Anthropometric Measurements

#### 2.2.1. Weight

A bench scale was used to weigh all of the participants (Model A1ZE, East Rand; maximum weight limit 300 kg calibrated electronic scales). The scale was leveled with the help of its inbuilt spirit level on an even, uncarpeted surface (if a zero (0) appeared in the top left side of the display window, the scale was level). All participants were weighed in the clinics and were instructed to remove shoes and be dressed in light clothing when their weights were recorded. Participants were instructed to step onto the scale, stand still and upright in the center of the platform, face the clinic assistant, and look straight ahead with their feet flat and slightly apart until the reading was taken. The reading was then entered into the section provided on the clinical assessment form by the clinic assistant. Weight was measured to the closest hundredth of a gram. The individual was asked to take a step back from the scale. The clinic assistant waited for the zero reading to appear on the digital display after the participant stepped down from the scale before repeating the procedure. The difference between the 2 values had to be less than 100 g. If not, the scale was examined for accuracy, and the process had to be repeated until the two readings were within 100 g. Both measures were recorded on the clinical examination form. A third measurement was conducted if the two measures differed by more than 100 g. For inspection, the two measurements that were closest to each other were chosen.

#### 2.2.2. Height

The standing height was taken using a stadiometer (Seca Model 213; Medical Scales and Measuring Systems). The participant’s shoes were removed and the stadiometer was placed on an even, uncarpeted surface. If the participant had his hair tied on top of his head, it was untied, and he was aligned in front of the clinic assistant, facing directly ahead with his head in the Frankfort plane. Shoulder blades, buttocks, and heels lightly touched the stadiometer’s stand, arms comfortable at sides, legs straight with knees together, and feet flat with heels touching. The measurement was recorded on the clinical examination, and the procedure was repeated once more. A total of two readings were recorded. If the two readings differed by more than 0.1 cm, a third measurement was taken. For further examination, the two measurements that were closest to each other were utilized.

#### 2.2.3. Body Mass Index

BMI was calculated for all participants using the equation: weight (kg)/height (m^2^) and the recommended WHO cut-off points were used to determine normal weight (BMI = 18.5–24.9) and overweight/obesity (BMI ≥ 25) [30].

#### 2.2.4. Waist Circumference

The individual stood upright/erect, abdominal relaxed, arms at sides, feet together, and weight evenly distributed between both legs. In the mid-axillary line, the lowest rib-margin and the iliac crest were located. The midpoint between the two anatomical landmarks was used to determine the waist circumference level, which was measured by wrapping a non-stretch fiberglass tape (Seca Model 203) horizontally around the abdomen. To avoid clenching their muscles or holding their breath, participants were instructed to breathe normally and lightly while the measurement was taken. The measurement was taken without squeezing the skin with the tape and was taken down to the nearest 0.1 cm. Two measures were obtained and entered on the clinical examination form. A third measurement was taken if the two measures differed by more than 0.1 cm. For inspection, the two measurements that were closest to each other were chosen [31,32]. A waist circumference ≥94 cm in men indicated central obesity [33].

#### 2.2.5. Hip Circumference

Participants stood erect and aligned themselves similarly to how they did for the waist circumference measurement, with arms at their sides and feet slightly apart. The measurement was obtained at the place where the circumference of the buttocks reached its maximum [32] with the non-stretch tape held in a horizontal plane, touching the skin but not indenting the soft tissue [31]. Measurement was obtained to the nearest 0.1 cm. Two measures were obtained and entered on the clinical examination form. A third measurement was taken if the two measures differed by more than 0.1 cm. For additional investigation, the two measurements that were closest to each other were chosen.

#### 2.2.6. Waist-to-Hip Ratio

By dividing the waist circumference by the hip circumference, the waist-to-hip ratio was calculated. A value > 0.91 for men was regarded as indicative of central obesity [33].

#### 2.2.7. Waist-to-Height Ratio

The waist-to-height ratio was calculated as the ratio of waist-to-height by dividing the waist circumference by the height. Central obesity was computed as WHtR > 0.5 [33].

Training was led by a certified anthropometrist, who was assisted along other personnel who had prior experience in taking anthropometric measurements. Trained survey staff conducted the actual measurements.

### 2.3. Biomarkers

#### Glycated Hemoglobin

Nurses and doctors in the clinic took a blood sample from the antecubital fossa. In adults, approximately 15–20 mL of blood was collected. Only consenting household members’ blood samples were collected, aliquoted into appropriate blood specimen collection tubes, mixed as needed, kept in a cooler box with ice packs, and couriered daily to the designated laboratories within 24 h of the time a blood sample was collected. The appointed laboratories were Pathcare and Lancet Laboratories. Both entities are South African National Accreditation System (SANAS) accredited. The biomarker studies were carried out using automated techniques such as high-performance liquid chromatography (HPLC) (HbA1c). Deviations from specified internal and external quality control methods have to be notified in accordance with the standard. There were none reported. The coefficient of variation for the analyses ranged from 0.5 to 3.75%, according to the analytical quality control documentation.

A threshold of 6.5% for HbA1c was used for the diagnosis of DM in the current analysis [34,35]. Participants were considered to have DM if they had HbA1c levels greater than or equal to 6.5% or were currently taking either oral glycemic medication or insulin.

### 2.4. Statistical Analysis

Data for males aged 20 years and older who had completed anthropometric assessments and had their HbA1c measured (N = 1 405) were analyzed. Continuous variables were presented as the means and standard deviations (M ± SD). The t-test was used to compare the means of the groups. The Chi-square and trend tests were used to analyze categorical variables, which were reported as numbers and percentages. The data were stratified by age group categories as follows: 20–44 (*n* = 695) and over 45 years (*n* = 710). The screening ability of anthropometric indices (BMI, WC, WHR and WHtR) to identify individuals with DM was explored using the ROC analysis. Plots of sensitivity versus 1 minus specificity were constructed for each of the indices. The AUC of the ROC and 95% confidence intervals (CIs) were used to identify which indicator had the best DM screening accuracy. The AUC is a measure of discrimination, and an AUC of 0.5, 0.6 ≤ AUC < 0.7, 0.7 ≤ AUC < 0.8, 0.8 ≤ AUC < 0.9, and ≥0.9 corresponds to no discrimination, poor, acceptable, excellent, and outstanding discrimination, respectively [36]. The maximum value of Youden’s index, calculated as sensitivity + specificity − 1, was used to determine the optimal cut-off point for each index to identify individuals with DM [37]. A logistic regression analysis, fitted by age group, race, employment, province, locality, education, triglycerides, LDL-C and total cholesterol using multivariable fractional polynomials (MFP) and reporting odds ratios (OR) was used to measure the association of each of the indices (WHR, WHtR, WC and BMI) with the odds of having DM. Three logistic regression models were applied: model 1, adjusted for age group; model 2, adjusted for age group, race, employment, province, locality, and education; and model 3, further adjusted for age group, race, employment, province, locality, education, triglycerides, LDL-C and total cholesterol. Data were analyzed using Statistical Program for Social Sciences (SPSS, version 25.0, Chicago, IL, USA). All statistical tests were two sided, and differences were considered statistically significant at *p*-values < 0.05. Because of SANHANES-1’s multistage cluster sampling design, some individuals had a higher or lower probability of selection than others. Estimates may be skewed as a result of the unequal sampling. Sample weights were introduced to correct for bias at the EA, household, and individual levels, as well as to adjust for non-response. EA sampling weights were calculated when drawing the 500 EAs. These EA sampling weights were calculated to account for unequal size measurements during sampling. However, not all 500 EAs were carried out. As a result, the sampling weights for these EAs were adjusted for non-response at the EA level. Furthermore, because not all targeted VPs were realized, VP sampling weights were calculated based on realized and valid households. Demographic, physical examination, and clinical examination data on all individuals in all households in all responding EAs were then compiled in order to calculate individual sample weights at each responding level (questionnaire, physical and clinical examination). This weight was calculated by multiplying the final VP sampling weights by the selected person’s sampling weight per VP and age group. This procedure yields a final sample that is representative of the South African population in terms of gender, age, race, locality type, and province. The survey was intended to be generalizable to the entire South African household population. The weighting procedure described here was carried out with SAS version 9.3 and the CALMAR macro for benchmarking.

## 3. Results

### Descriptive Analysis

Table 1 presents the socio-demographic characteristics, anthropometrical indices as well as the HbA1c levels of the participants. There was almost an equal spread of participants in both age groups (i.e., 49.5% and 50.5% in age groups 20 to 44 and >45 years, respectively). The majority (63.5%) of the participants were of Black African descent and only 36.5% were non-Black. The majority (53.0%) of the participants resided in urban formal settlements. A total of 34.6%, 10.5%, 10.4%, 36.6% and 61.0% of the participants were overweight and obese based on their BMI levels and had abnormal HbA1c, WC, WHR and WHtR, respectively.

Table 2, on the other hand, shows that the mean values for BMI, WC, WHR, WHtR and HbA1c were 24.1 kg/m^2^, 83.0 cm, 0.9, 0.5 and 5.9%, respectively.

Table 3 shows the risk (as shown by ORs) for DM as predicted by BMI, WC, WHtR and WHR of the participants. Overall, there were significant associations (all *p* values were < 0.05) between all anthropometric indices and DM before and after the data were adjusted. Before adjusting, it was shown that participants who had higher than normal BMI and abnormal WC, WHR and WHtR were 5-, 7-, 7- and 12-fold more likely to have higher than normal levels of HbA1c. In this case, the ORs were 5.06 at 95% CI: 3.47–7.37, 7.13 at 95% CI: 4.78–10.65, 7.06 at 95% CI: 4.70–10.62, and 12.15 at 95% CI: 5.63–26.22, respectively, with all *p* values < 0.001. After removing the confounding effects of age group, ORs decreased slightly for BMI and WC to 4.14 at 95% CI: 2.81–6.10 and 6.29 at 95% CI: 4.14–9.55, respectively; and decreased substantially for WHR and WHtR to 4.80 at 95% CI: 3.14–7.33 and 8.11 at 95% CI: 3.72–17.67), while all the *p* values remained < 0.001. On further removal of the confounding effects of age, race, employment, province, locality, education, triglycerides, LDL-C and total cholesterol, ORs decreased for all indices to 2.445 at 95% CI:1.213–4.929; *p* = 0.012 for BMI, 4.950 at 95% CI: 2.243–10.926; *p* < 0.001 for WC, 2.926 at 95% CI: 1.503–5.697; *p* = 0.002 for WHR, and 4.590 at 95% CI: 1.603–13.141; *p* = 0.005 for WHtR. It is also important to note that ORs for BMI were the lowest both before and after adjusted ORs.

The ROC analysis outcomes for WC, WHR and WHtR exhibited excellent abilities to predict DM (i.e., as all outcomes were above 80%). For instance, the AUC for WC was 80.4%, 80.2% for WHR and 80.6% for WHtR (Figure 2a and Table 4). The cut-off points to predict DM for WC, WHR and WHtR were calculated to be ≥88.95 cm; >0.92 and >0.54, respectively. The sensitivity and 1-specificity for WC were 71.0% and 26.2%, 70.3% and 25.7% for WHR, and 70.3% and 21.1% for WHtR, respectively. The ROC analysis outcome for BMI, on the other hand, only exhibited an acceptable ability to predict DM. The AUC for BMI in this case was less than 80% (i.e., it was equal to 75.6%), with the cut-off point calculated to be ≥24.64 kg/m^2^ and the sensitivity and 1-specificity being 70.3% and 31.9%. Despite the AUC for BMI being less than the AUCs for WC, WHR and WHtR, based on the overlapping CIs, no significant differences were observed.

Because our findings in Table 3 showed that age influenced the interaction between the anthropometric indices and the DM, we decided to stratify the ROC curve analysis by age group (Figure 2b,c and Table 5) to determine which indices predicted the risk of DM better in the younger (20–44 years) or the older (>45 years) groups of South African men. In this case, the AUCs for all indices became lower than 80%, showing an only acceptable ability to predict the risk of DM. Of note is that, while no significant differences were observed based in the overlapping CIs, the AUCs showed that WC and WHtR performed better in predicting the risk of DM in the older age group (with AUCs of 75.4% and 75.3%), with WHtR being the only index that performed better in the younger age group (AUC of 78.8%). On the other hand, all the cut-off points to predict DM were higher in the older age group compared to the younger group (i.e., were ≥24.7 kg/m^2^, ≥89.8 cm, >0.93 and >0.54 versus ≥24.2 kg/m^2^, ≥87.5 cm >0.88 and >0.48 for BMI, WC, WHR and WHtR, respectively). As for the sensitivity values, BMI and WC values were higher in the older age group (i.e., were 70.7% and 71.5%) and WHR and WHtR values were higher in the younger age group (i.e., both were equal to 73.3%).

## 4. Discussion

The current study sought to assess the sensitivity and specificity of BMI to predict DM as measured by HbA1c in South African males aged 20 years and older, and compare the performance of BMI against the performance of other anthropometric indices (i.e., WC, WHR and WHtR). The notable outcomes were that, based on the AUCs, BMI showed acceptable ability to predict the risk of DM in South African males aged 20 years and older. Moreover, no significant differences were observed between the AUC for BMI and the AUCs for WC, WHR, and WHtR. However, the AUCs for WC, WHR, and WHtR were above 80%, an indication that they were stronger predictors of DM. We can explain the inferior performance of BMI to predict DM by the fact that BMI is not sensitive to body fat distribution, especially central obesity, a condition that is observed in the majority of older South African men [38,39]. On the other hand, we can attribute the excellent performance of WC, WHR and WHtR to the fact that they measured central adiposity [7,8,9,10], hence they performed better in the current group of participants.

Moreover, our current outcome that suggested age to be a confounder when using anthropometric indices to predict the risk of DM is also corroborated by other similar international studies [26,27,40,41]. For instance, Chen et al. [27] and Cheng et al. [42] have shown that age mediates the ability of these anthropometric indices to predict metabolic syndrome (MetS), and its components such as DM and hypertension. For this reason, in the current study, participants were stratified into two age groups. Despite AUCs lower than the outcomes of the overall analysis, the AUCs showed that WC and WHtR still performed better in predicting the risk of DM in older men (the AUCs were both above 75%), while WHtR even performed close to excellent in predicting the risk of DM in younger men (AUC = 79%). These outcomes are in contrast to those reported in other international studies, where anthropometric indices such as WHtR are shown to have stronger associations with cardiometabolic risk factors including DM in middle- to older-adult population groups, compared to the younger adults [43,44]. In these studies, such differences were attributed to fat mass gain and lean mass loss at old age [45,46]. However, we acknowledge that participants in these studies were stratified into three and more age groups, while we only stratified the participants into two age groups in our study.

Our outcomes also magnified the mediocre performance of WHR to predict diabetes in men, when compared to WC and WHtR. We can interpret these outcomes using other international studies that suggest WC to be an accurate and simple measure of abdominal obesity as compared to WHR [47], while WHtR is regarded as a measure that takes height into account when predicting central adiposity [46,48]. According to the aforementioned studies, height is a more sensitive indicator to muscle mass distribution throughout the entire body than the hip circumference used in WHR. Generally, men do not have big hips [49]; and in South Africa, evidence suggests that younger men are taller and leaner (i.e., masculine) than older men [39]. In fact, muscle mass is replaced by fat mass especially around the waist in older adult men [39]. Moreover, shorter stature and higher waist circumference translate to higher WHtR outcomes [50]. According to the outcomes of the current study, WHtR increased with an increase in the HbA1c, where older men had higher HbA1c than younger men (data shown elsewhere [2,11]). However, we cannot discount other risky behaviors (i.e., smoking or drinking) that men engage in, that have been shown to contribute to the increase in HbA1c outcomes especially in the majority of older South African men [51].

The outcomes from the current study and other recent systematic reviews [44,52] also highlighted that WHtR is a better predictor for cardiometabolic risk factors, including DM when compared to BMI and WC alone. Based on the outcomes of the current analysis, it is also shown that WC, WHR and WHtR can excellently and individually predict the risk of DM. This outcome is consistent with other international literature that advocate the individual use of WC, WHR or WHtR [5,26,27]. However, pairing BMI with WHtR, on the other hand, could leverage its performance, especially in South African surveillance studies of men to predict DM. This is because, in addition to determining the overall adiposity (as measured by BMI), WHtR will also give an indication of where fat is stored in the body, whether in the visceral or abdominal region relative to the gluteal region [53]. WHtR will also give an added indication of the role played by the height of an individual in specifying central obesity, especially in a country such as South Africa that grapples with stunting even in adult men [1,2].

Finally, while the World Health Organization (WHO) [33] and the International Diabetes Federation (IDF) [12] recommend the use of pre-specified cut-off points for BMI, WC, WHR and WHtR to standardize comparisons within and between populations [33,54,55], such cut-off points are based on research centering European, Asian, Chinese and Japanese people, and may not apply to other ethnic groups, especially those of African descent—there is substantiated evidence to suggests that the current WHO [33,55] and IDF [12] cut-off points slightly underestimate the screening of DM among other ethnicities, especially in men. Our current outcomes may therefore mitigate this shortfall in that it may be necessary to lower all the cut-off points to those obtained for younger men (i.e., ≥24.23 kg/m^2^, ≥87.45 cm, >0.88 and 0.48 for BMI, WC, WHR, and WHtR, respectively). Doing this may improve the predictability of DM in adult South African men regardless of age. These outcomes could also be applied to other adult men of African descent in the southern region of Africa, who have similar characteristics as those in South Africa.

While the current study have many strengths, there are limitations that need to be taken into consideration when interpreting the outcomes. Firstly, we could not demonstrate the causal relationship between anthropometric indices and DM because of the cross-sectional nature of our study design. As such, only associations were measured and reported. Secondly, this study focused only on male participants, thus limiting gender difference comparisons. Our focus was on men because there is substantiated evidence from South African to suggest that the prevalence of uncontrolled DM is higher in females and is mediated by their large body size as measured by BMI. Based on this background, we became concerned that uncontrolled DM may be underestimated in South African men. This could be partly due to the existing evidence that unlike South African women who are obviously obese, the mean BMI for South African men is still regarded to be within the normal range of weight (between 22.5 and 24.5 kg/m^2^), because few of them present with overweight and obesity [56]. Normal-weight individuals are often assumed to be free of diseases if they do not present with symptoms for these diseases, hence they are often overlooked when implementing targeted interventions to mitigate metabolic diseases such as DM. Moreover, unlike women, South African men are reluctant to attend health services/or participate in health screening/surveillance activities [11]. Hence, the majority of them only present to health service centers when their DM condition has progressed. This therefore results in them having a poor prognosis for this disease. We have also already highlighted our concern in the introduction that, if not tightly monitored, the proportion of South African men who are living with DM may surpass that of South African women in the future, since current predictions suggest that there are already more males than females (66% versus 64%) who are pre-diabetic in South Africa [1]. Further, we are conducting similar research in South African women, and this analysis is ongoing. Finally, we did not differentiate between type 1 or type 2 DM as this was beyond the scope of the current research.

## 5. Conclusions

To conclude, we packaged the notable outcomes from this research in Box 1 below. These suggest that both overall adiposity (as shown by BMI) and abdominal adiposity (as shown by WC, WHR and WHtR) play an important role in predicting the risk of uncontrolled DM (as measured by HbA1c) in men. However, not all anthropometric indices have the same performance in predicting the risk for DM. For instance, indices that consider fat deposition, especially around the waist, such as WC, WHR and WHtR, show excellent performance in predicting DM, while BMI, which is not sensitive to body fat distribution, only shows acceptable ability to predict DM. Despite the inferior performance of BMI, our findings show that BMI could still be an acceptable indicator to identify South African men who are at risk of having DM since there are no significant differences between the performance of BMI and other anthropometrical indices. Our high AUCs, show that WC, WHR and WHtR could be used independently to predict the risk of uncontrolled DM in South African men. However, if researchers are interested in predicting the maximum number of South African men at risk of this disease, regardless of age, BMI could be paired with WHtR. Waist-to-height ratio outcomes will also identify stunted men with a bigger waist circumference who may be missed by BMI cut-off points. Finally, because we showed that the interaction between the anthropometric measurements included in the current research and DM was mediated by age, we recommend the use of lower cut-off points than those pre-specified by the WHO and the IDF in order to improve the predictability of DM in all age groups of adult South African men. These cut-off points are for younger men and are presented in Table 5 (i.e., ≥24.23 kg/m^2^, ≥87.45 cm, >0.88 and 0.48 for BMI, WC, WHR, and WHtR, respectively). We therefore think these current outcomes are of importance as they may aid in health promotion directed at improving the nutritional status of South African men. They can also be used to improve DM surveillance in the country, more especially in screening adult men of all age groups who may have uncontrolled DM, as well as support targeted interventions to control DM in a country such as South Africa that grapples with metabolic disorders.

Box 1Take-home messages from the current research.Based on the current research, 34.6%, 10.5%, 10.4%, 36.6% and 61.0% of South African males aged 20 years and older had BMI ≥ 25 kg/m^2^, HbA1c ≥ 6.5, WC ≥ 94, WHR > 0.91 and WHtR > 0.5, respectively.After adjusting for age group, South African men with abnormal BMI, WC, WHR and WHtR were 4-, 6-, 5-, and 8-fold more likely to present with higher abnormal levels of HbA1c.Further adjusted for age group, race, employment, province, locality, education, triglycerides, LDL-C and total cholesterol, ORs decreased for all indices to 2 for BMI, 5 for WC, 3 for WHR, and 5 for WHtR to pre-sent participants with higher abnormal levels of HbA1c.Based on the area under the curve (AUC) outcomes, WC, WHR and WHtR excellently predicted the risk of DM (with corresponding AUCs of 80.4%, 80.2% and 80.6%, respectively).
○This means that these indices could be used independently to predict the risk of DM.
Body mass index (BMI) shows acceptable ability to predict the risk of DM (i.e., AUC of 75.6%).
○This means that BMI could still be used independently to predict the risk of DM.○However, we recommend pairing it with another strong index (especially a high-performing index such as WHtR) that considers central adiposity to supplement its ability to predict the risk of DM.
Based on confidence interval (CI) levels that do not overlap, the AUC for BMI was not significantly differ-ent from those of WC, WHR and WHtR.Because age is a confounder when using anthropometric indices to predict the risk of DM, we recommend the use of lower cut-off points than those pre-specified by the WHO and the IDF, in order to improve the predictability of DM in all age groups of adult South African men.
○The following cut-off points are for younger men and are presented in Table 5 (i.e., ≥24.23 kg/m^2^, ≥87.45 cm, >0.88 and 0.48 for BMI, WC, WHR, and WHtR, respectively)


## Figures and Tables

**Figure 1 ijerph-19-03224-f001:**
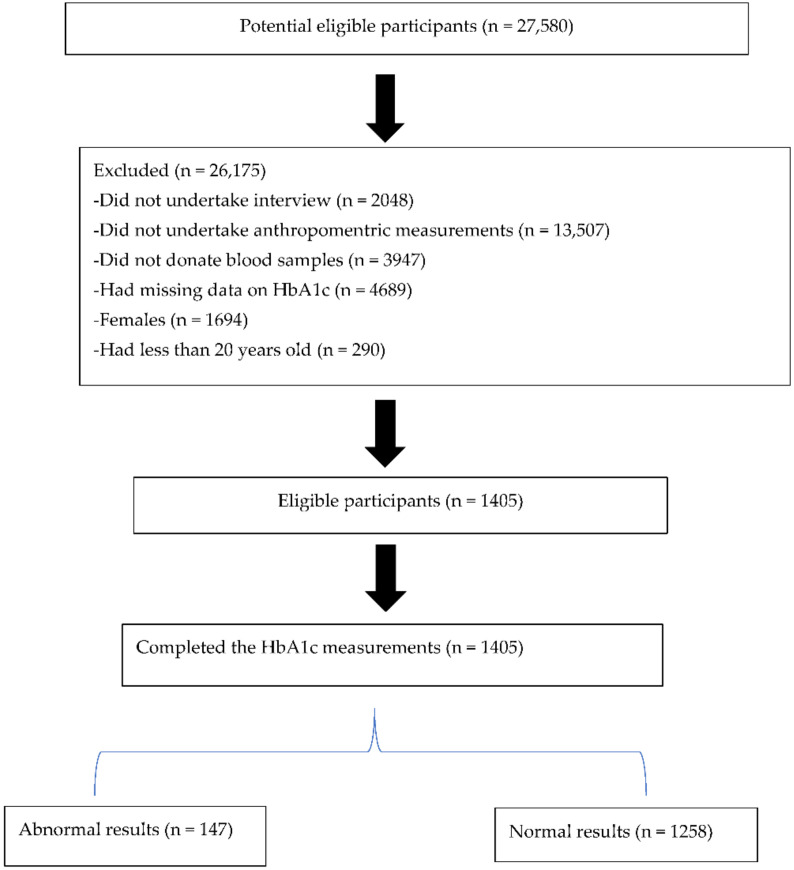
Flow diagram of subject selection for this study.

**Figure 2 ijerph-19-03224-f002:**
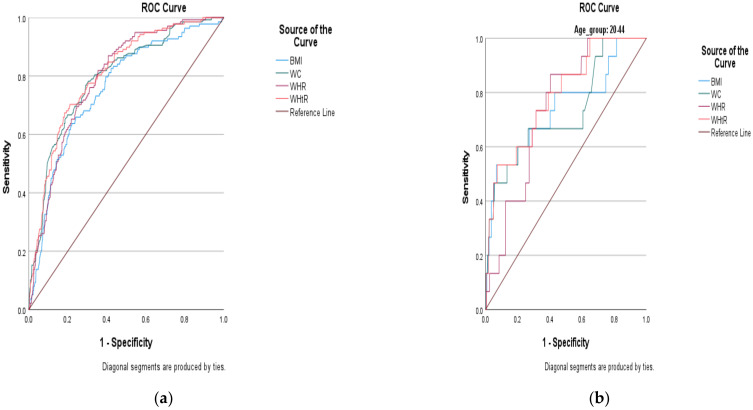
(**a**) ROC curves and optimal cut-off values for anthropometric indices in the prediction of diabetes mellitus in South African males aged 20 years and older. (**b**) ROC curves and optimal cut-off values for anthropometric indices in the prediction of diabetes mellitus in South African males who are 20–44 years old. (**c**) ROC curves and optimal cut-off values for anthropometric indices in the prediction of diabetes mellitus in South African males who are >45 years old.

**Table 1 ijerph-19-03224-t001:** Socio-demographic characteristics, anthropometric indices and HbA1c outcome of South African males aged 20 years and older: SANHANES-1.

	*n* (%)
Age Group (Years)	
20–44	695 (49.5)
>45	710 (50.5)
Race ***	
Black *	888 (63.2)
Non-Black **	510 (36.3)
Locality	
Urban formal	744 (53.0)
Urban informal	147 (10.5)
Rural formal (farms)	260 (18.5)
Rural informal (tribal)	254 (18.1)
Body mass index (kg/m^2^)	
Normal BMI, 18.5–24.9	878 (65.4)
Overweight/obesity, >25	465 (34.6)
Diabetes mellitus (%)	
Normal, HbA1c < 6.5	1258 (89.5)
Abnormal, HbA1c ≥ 6.5	147 (10.5)
Waist circumference (cm)	
Normal, WC < 94	1203 (89.2)
Abnormal, WC ≥ 94	146 (10.4)
Waist-to-hip ratio	
Normal, WHR < 0.91	850 (63.4)
Abnormal, WHR > 0.91	491 (36.6)
Waist-to-height ratio	
Normal, WHtR < 0.5	470 (35.4)
Abnormal, WHtR > 0.5	857 (61.0)

* Black African Descent; ** Mixed Race, European Descent and Asian; Race *** = data have 7 missing values on race variable, and total percentage does not add up to 100%. BMI = body mass index, WC = waist circumference, WHR = waist-to-hip and WHtR = waist-to-height ratio.

**Table 2 ijerph-19-03224-t002:** The physiological characteristics of South African males aged 20 years and older.

Anthropometric Indices and HbA1c	Mean ± SD
Weight (kg)	67.3 ± 16.4
Height (cm)	167.9 ± 8.2
Body mass index (kg/m^2^)	24.1 ± 5.9
Waist circumference (cm)	83.0 ± 14.2
Hip circumference (cm)	93.9 ± 11.8
Waist-to-hip ratio	0.9 ± 0.1
Waist-to-height ratio	0.5 ± 0.1
HbA1c (%)	5.9 ± 1.0

**Table 3 ijerph-19-03224-t003:** The risk for diabetes mellitus among South African males aged 20 years and older by anthropometric indices: SANHANES-1.

	Unadjusted			Adjusted OR Model 1			Adjusted OR Model 2			Adjusted OR Model 3		
	Crude OR	95% CI	*p*-Value	AOR	95% CI	*p*-Value	AOR	95%CI	*p*-Value	AOR	95% CI	*p*-Value
BMI	5.061	3.474–7.374	<0.001	4.142	2.814–6.097	<0.001	3.687	2.260–6.016	<0.001	2.445	1.213–4.929	0.012
WC	7.133	4.779–10.647	<0.001	6.285	4.136–9.550	<0.001	6.533	3.746–11.394	<0.001	4.950	2.243–10.926	<0.001
WHR	7.064	4.698–10.623	<0.001	4.800	3.141–7.334	<0.001	4.836	2.881–8.118	<0.001	2.926	1.503–5.697	0.002
WHtR	12.151	5.632–26.215	<0.001	8.108	3.721–17.667	<0.001	8.406	3.235–21.840	<0.001	4.590	1.603–13.141	0.005

Model 1 = adjusted OR for age group. Model 2 = adjusted OR for age, race, employment, province, locality, and education. Model 3 = adjusted OR for age, race, employment, province, locality, and education, triglycerides, LDL-C and total cholesterol. BMI = body mass index, WC = waist circumference, WHR = waist-to-hip ratio, WHtR = waist-to-height ratio, and OR = odds ratio.

**Table 4 ijerph-19-03224-t004:** Outcomes that show the power of the anthropometric indices to predict diabetes mellitus: the area under the curve, sensitivity, 1-specificity and 95% confidence intervals.

Anthropometric Index	AUC	*p*-Value	95% CI	Cut-Off Point	Sensitivity	1-Specificity
BMI kg/m^2^	0.756	<0.001	0.714–0.798	24.64	0.703	0.319
WC cm	0.804	<0.001	0.754–0.833	88.95	0.710	0.262
WHR	0.802	<0.001	0.757–0.827	0.921	0.703	0.257
WHtR	0.806	<0.001	0.769–0.842	0.543	0.703	0.211

AUC = area under curve, BMI = body mass index, WHR = waist-to-hip ratio, WHtR = waist-to-height ratio, and WC = waist circumference.

**Table 5 ijerph-19-03224-t005:** Area under the curve and optimal cut-off points for anthropometric indices to predict diabetic mellitus: stratified by age group.

Age 20–44 Years (*n* = 695)								Age > 45 years (*n* = 710)				
Anthropometric Index	AUC	95% CI	*p*-Value	Cut-Off Point	Sensitivity	1-Specificity	AUC	95% CI	*p*-Value	Cut-Off Point	Sensitivity	1-Specificity
BMI kg/m^2^	0.742	0.593–0.890	0.001	24.23	0.667	0.271	0.724	0.676–0.772	<0.001	24.65	0.707	0.400
WC cm	0.729	0.583–0.874	0.002	87.45	0.600	0.201	0.754	0.707–0.801	<0.001	89.75	0.715	0.330
WHR	0.740	0.645–0.836	0.001	0.875	0.733	0.314	0.733	0.688–0.779	<0.001	0.926	0.707	0.358
WHtR	0.788	0.674–0.903	<0.001	0.483	0.733	0.314	0.753	0.707–0.800	<0.001	0.544	0.707	0.300

## Data Availability

The data presented in this study are available on request from the corresponding author. The SANHANES data are available on request from http://datacuration.hsrc.ac.za/ (accessed on 15 June 2019).

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
