# Peer review of "Comparison of the Ability of Anthropometric Indices to Predict the Risk of Diabetes Mellitus in South African Males: SANHANES-1"

_ijerph, 2022, doi:10.3390/ijerph19063224_

Round 1

Reviewer 1 Report

This Cross-sectional study aims to assess the sensitivity of body mass index (BMI) to predict the risk of diabetes mellitus (DM) and whether waist circumference (WC), waist-to-hip (WHR) and waist-to- 15 height (WHtR) ratios are better predictors of the risk of DM than BMI.

This paper is based on survey undertaken in 2012, to determine the nutrition and health status of the South African population.

It is a formally well conducted study, quite interesting in for the clinical practice. 

There are only a few minor remarks to be made to improve the paper:
1. Some parts such as the abstract and the conclusion are not very readable and difficult to interpret for a non-practitioner. in particular, the conclusions are well written (the box with the take-home messages is excellent) but need to be simplified.

2. The most common index used to assess a subject's metabolic risk is BMI combined with waist circumference. Based on the results of your paper, what suggestions can we give to a prevention practitioner to assess in a simple and cost-effective way to facilitate his work? Are BMI and anthropometric measurements enought? Is it necessary to prescibe a OGTT or assess glycosylated haemoglobin?

3.   Be careful because some paragraphs seem to be plagiarised.

Reviewer 2 Report

The authors analysed data of 1404 men collected from the  first South African National Health and Nutrition Examination Survey  (SANHANES-1) in 2012 to determine optimal cut-points of common anthropometric indices to predict diabetes and compare their ability to predict diabetes mellitus in South Africa. The study show that optimal cut-points of BMI, WC, WHR and WHtR for diabetes mellitus were 24.64 kg/m2, 88.95 cm, 0.921 and 0.543. It also concluded that WHtR was the best indicator to predict diabetes mellitus in this population. My major concern is the lack of validation of the identified cut-points. The authors should undertake internal validation to consolidate your findings, using some common methods such as cross-validation, bootstrapping. Other comments are as follows:

  • I was wondering why only men were included in this study
  • Have the authors considered sample size estimation for this study and how?
  • This study used "a stratified cluster sampling approach with multiple stages" but I did not see the authors reported they have accounted for the clustering of participants. If so, please describe your analytical approaches to a complex survey in Statistical section.
  • I was wondering why only age was adjusted for in logistic regression analyses, while multiple factors know to affect diabetes mellitus were collected in the survey.

Reviewer 3 Report

Comparison of anthropometric indices

Prediction of risk of diabetes is important in a world of escalating Obesity and escalating incidence of diabetes. This important paper investigates the sensitivity of BMI in a South African population and compares BMI to Waist circumference, Waist to hip and waist to height ratios as predictors of diabetes. There is an excellent introduction which is well referenced.

The Materials and methods section concise and well written. The results well displayed and the discussion well structured. The study only done in men. The power of BMI to predict diabetes varies in different populations and I was surprised to read the BMI is an excellent predictor of diabetes in this population.The Authors found that the other 3 measurements tested performed slightly better but perhaps only marginally so.

A very interesting study in this South Africian male population

Round 2

Reviewer 2 Report

The authors have made great efforts in addressing my previous comments; however, there are a few things that I would like you to explain and/or add to the revised version:

  • Age is an non-modifiable factor and well known predictor for diabetes mellitus, so that's good for you to adjust for it. Nonetheless, there are other factors collected from this study that have been found to predict diabetes mellitus. I would suggest you account for those variables in multivariable logistic regression to strengthen your results.
  • The authors stated that age mediated the association between anthropometric indices and diabetes. What data did you base on and possible explanations in terms of biological pathways? The threshold of 45 year was used to conduct stratified analyses. Was it chosen based on your data distribution or literature (if so, please add a suitable reference). If the authors thought age was an effect modifier, an appropriate test is necessary to make conclusion.
  • The authors can improve reporting results of diagnostic tests in reference to the most recent STARD guideline (https://bmjopen.bmj.com/content/6/11/e012799). For example, the authors did not report how measures of diagnostic accuracy were compared and statistical tests used for such comparison. Or inclusion and exclusion criteria were used or how missing data were handled. It would be more informative to provide a flow diagram of subject selection for this study.
  • The authors asserted that other parameters such as false-negative and false-positive rates were computed but I did not see their results.

Author Response

This manuscript is a resubmission of an earlier submission. The following is a list of the peer review reports and author responses from that submission.

Round 1

Reviewer 1 Report

This is a cross-sectional study in which possible correlations between anthropometric parameters and risk of diabetes mellitus were evaluated. 
The study was conducted well, although with all the limitations of being retrospective (data from 2012) and excluding female subjects.
The discussion needs to be improved.
As a study suggesting practical and useful aspects for everyday clinical practice it would be useful to include more precise "take home messages". In the light of the data obtained, what suggestions can we make to the clinician? For which data (anthropometric values, age, etc.) is it advisable to prescribe the glycated haemoglobin dosage? A summary table would be helpful. 

Be careful because there are whole paragraphs plagiarised (see attached report). Please rewrite those sentences.

Reviewer 2 Report

-p-values cannot be 0.000. Update the p-values of 0.000 throughout the paper.

-Line 23-25, it is the data based on unadjusted analysis. The relative risk for diabetes was also measured after adjusting for age. Therefore, the current sentence should be replaced with the age-adjusted relative risk for diabetes

- The study included males aged 15-21 years. For their BMI, either BMI for age percentile or BMI z- score should be used. Therefore, their BMI should not be combined with adults’ BMI.

-Line 180, waist-to-hip ratio > 0.91 for men was selected in a cross-sectional study with adults aged the same as or greater than 20 years. Therefore, a different ratio cutoff should have been used for males aged 15-19 years.

-Table 1, race: a total of black and non-black is 1143 although 1694 males were included in the study.

-Table 1, the mean waist-to-hip ratio should be presented up to two decimal points since the cutoff is 0.91.

-Table 3, check the following data: unadjusted 95% confidence interval for waist-to-height ratio, age-adjusted 95% confidence intervals for waist-to-hip ratio and waist-to-height ratio.

Lines 327-333 are redundant to Lines 303-308.

-Check a list of references: missing journal names, volume numbers, and page numbers

Reviewer 3 Report

This is a well-written manuscript. Although the author mentioned that the risk factors for diabetes in men and women are different. However, the author's reason for showing only results in men is strange. I suggest that authors should also provide women's results in the same paper.

Minor comment:

  1. Prevent repeating the results of tables in the main text.
  2. Improve the quality of Figure 1. What’s “ties”?
  3. Line 368, typo “out come”